

# Predicting Sea Surface Temperatures with Coupled Reservoir Computers

Benjamin Walleshauser[1,3] and Erik Bollt[2,3]

[1]Department of Physics and Department of Mechanical Engineering, Clarkson University
[2]Department of Electrical and Computer Engineering, Clarkson University
[3]Clarkson Center for Complex Systems Science

**Correspondence:** Benjamin Walleshauser (wallesbt@clarkson.edu)

**Abstract. Sea surface temperature (SST) is a key factor in understanding the greater climate of the Earth and is an important variable when making weather predictions. Methods of machine learning have become ever more present and important in data-driven science and engineering including in important areas for Earth Science. We propose here an efficient framework that allows us to make global SST forecasts by use of a coupled reservoir computer method**
**that we have specialized to this domain allowing for template regions that accommodate irregular coastlines. Reservoir computing is an especially good method for forecasting spatiotemporally complex dynamical systems, as it is a machine learning method that despite many randomly selected weights, it is nonetheless highly accurate and easy to train. Our approach provides the benefit of a simple and computationally efficient model that is able to predict sea surface temperatures across the entire Earth's oceans. The results are demonstrated to replicate the actual dynamics of the**
**system over a forecasting period of several weeks.**

## 1 Introduction

As most of Earth's surface is covered by water, global sea surface temperatures (SST) are an important parameter in understanding the greater climate of the Earth. Sea surface temperature (SST) is an important variable in the study of marine ecology (Gomez et al., 2020; Novi et al., 2021), weather prediction (Dado and Takahashi, 2017), and to help predict future climate
scenarios (Pastor, 2021). Yet the task of predicting changes in the SST is quite difficult, due to large variations in heat flux, radiation, and diurnal wind near the surface of the sea (Patil et al., 2016).

Given the importance of sea surface temperature to the fuller Earth weather and climate system, there is significant interest in forecasting this spatiotemporally complex process. Methods for predicting changes in the SST can be divided into two different categories: numerical methods and data-driven methods. Numerical methods are based on the underlying knowledge
of the governing physics behind the system, and simulation thereof. These are widely used to predict SST over a large area (Stockdale et al., 2006; Krishnamurti et al., 2006). However, data-driven methods encompass statistical and machine learning approaches, and are widely used to predict the SST, often with little to no knowledge regarding relevant physical principles behind the dynamics of the system, hence reducing the complexity of the model. Several statistical methods that have been used include: Markov models (Xue and Leetmaa, 2000; Johnson et al., 2000), linear regression (Kug et al., 2004), and empir-





ical canonical correlation analysis (Collins et al., 2004). Meanwhile machine learning methods have included: support vector machines (SVM) (Lins et al., 2013), long short term memory (LSTMs) (Zhang et al., 2017; Kim et al., 2020; Xiao et al., 2019), memory graph convolutional networks (MGCNs) (Zhang et al., 2021), etc. In this paper, we utilize coupled reservoir computers (RC), therefore taking advantage of the reduced complexity of data-driven methods, while still being able to predict temperatures globally due to the minimal training required by each RC. We have adapted the RC concept for spatiotemporal

processes to allow for coupled local templates that accommodate the peculiarities associated with varying coastlines.

Reservoir computers have been shown to be excellent predictors of complex dynamical systems (Ghosh et al., 2021; Pandey and Schumacher, 2020) regardless of the relative simplicity of the approach. They have even been shown to be proficient in the prediction of spatiotemporally complex systems, such as the Kuramoto-Sivashinsky PDE (Vlachas et al., 2020) and cell segmentation (Hadaeghi et al., 2021). The use of coupling is introduced due to the large number of points on the map, making

it computationally challenging to utilize a single large reservoir computer. The reservoirs are coupled together by making the reservoirs functions of points outside their forecast domain, effectively creating overlap.

## 2 Background

Reservoir computing is a type of recurrent neural net where the only layer trained is the output layer, which is done with a simple linear method. Compared to traditional recurrent neural networks, reservoir computers utilize randomly generated input

and middle weights, which in effect reduces training time significantly. The reservoir computer is stated as follows (Jaeger and Haas, 2004),

$$\boldsymbol{r}_{i+1} = q(\mathbf{A}\boldsymbol{r}_i + \mathbf{W}_{in}\boldsymbol{X}_i + \boldsymbol{b}) \tag{1}$$

$$\boldsymbol{Y}_{i+1} = \mathbf{W}_{out}\boldsymbol{r}_{i+1} \tag{2}$$

The inputs $\boldsymbol{X}_i$ (of total length $dx$) is the raw data which describes a system, in our case these would be the temperatures

at points in the sea. These are fed into the reservoir via the input matrix $\mathbf{W}_{in}$, which has weights which are determined via sampling a uniform distribution $U(-\sigma, \sigma)$. The reservoir dimension $N$ describes how many nodes there are to be within the reservoir. Therefore in order to transform the inputs into the space of the reservoir, the dimensions of $\mathbf{W}_{in}$ will be $N \times dx$.

The reservoir state $\boldsymbol{r}$, which evolves according to Eq. 1, carries information about the current and previous states of the system. During training, the reservoir states are horizontally concatenated as time evolves to form the matrix $\mathbf{R} = \left[\boldsymbol{r}_1|\boldsymbol{r}_2|\ldots|\boldsymbol{r}_{t_{train}}\right]$,

where $t_{train}$ is the number of data points being used to train the model, and so correspondingly the computational complexity associated with the matrix operations stated in Eq. 2. The dimensions of $\mathbf{R}$ at the end of the training phase will then be $N \times t_{train}$.

The reservoir matrix $\mathbf{A}$ which contains the middle weights, is a sparse matrix with with a set density $d$, with nonzeros values that are sampled from a uniform distribution $U(-\beta, \beta)$. $\mathbf{A}$ will have dimensions of $N \times N$ corresponding to the $N$ reservoir





nodes. The spectral radius $\rho$ of $\mathbf{A}$ is an important metaparameter in the formation of the RC (Jiang and Lai, 2019), and can be adjusted by scaling $\mathbf{A}$ (which essentially just involves changing $\beta$). The activation function $q(\mathbf{s})$ is usually picked to be a nonlinear function such as the sigmoid function or the hyperbolic tangent function. Often, a bias term $b$ is included when one desires to shift the activation function a set amount. After the training data set is cycled through and the matrix $\mathbf{R}$ is completed, the output matrix $\mathbf{W}_{out}$ is then found via a ridge regression:

$$\mathbf{W}_{out} = \mathbf{Y}\mathbf{R}^{\mathbf{T}}(\mathbf{R}\mathbf{R}^{\mathbf{T}} + \lambda\mathbf{I})^{-1} \tag{3}$$

Which utilizes a regularization parameter $\lambda$ to prevent overfitting. As the output matrix is transforming reservoir states to a desired output $\boldsymbol{Y}_{i+1}$ (of length $px$), the dimension of $\mathbf{W}_{out}$ will be $px \times N$. The trained model can then be used to forecast autonomously by inserting the newly predicted values $\boldsymbol{Y}_{i+1}$ back into the reservoir on the next iteration as $\boldsymbol{X}_i$.

## 3 Coupled Reservoir Computers

As we would like to predict the SST across the entire globe, it is computationally challenging to use a single reservoir computer to forecast for the entire map due to the number of points on the map. From the data set, there will be a total of $n * m$ points on the map. If we were to use a single reservoir computer, the size of the input data $\boldsymbol{X}_i$ would be around $0.71 * n * m$ (as 71% of the Earth's surface is water) for each time step. It will be seen in the *Data* section that the values of $n$ and $m$ that are ultimately to train our model are 120 and 240 respectively, hence the total number of points that are not on land across the map is about

20448. Hence, even if the reservoir dimension $N$ is $15:1$ with respect to the size of the input data (which is very small from previous experience, for example the 3D Lorenz system requires a value of $N$ around 300), the size of the resulting matrix $\mathbf{A}$ would be $306720 \times 306720$, which assuming each element requires 8 bytes, would require 750GB to store.

Therefore in this application, it is better to follow the methodology laid out by Pathak et al. (2018) to model the evolution of the SST with the use of smaller reservoir computers which each cover a small domain of the map. This approach is advan-

tageous as the input and middle weights for a reservoir computer are chosen randomly, hence allowing us to reuse the same input matrices[1] $\mathbf{W}_{in}$ and the same reservoir matrix $\mathbf{A}$ between the individual RCs. Now, each individual RC learns the local behavior of the points within it's forecasting domain (defined as a *pack*) while still being connected to a the global system via coupling with it's neighbors.

A pack is essentially a collection of contiguous indices on the greater map that will be assigned to a given RC. In other

words, a pack is an RC model of the dynamics, for a local region of the globe, that is designed to accommodate any local peculiarities of the land-water interface and also it couples into other neighboring packs. This reservoir computer will then be solely responsible for predicting the temperatures within it's pack as time evolves. For simplicity's sake in this study, points within a pack are grouped in the shape of rectangles, where the number of rows of points within the pack is defined as $n_{pack}$ and the number of columns subsequently $m_{pack}$. Therefore, each RC will be responsible for forecasting a maximum of

---

[1]Due to varying values of dx due to points on land, we cannot share a single input matrix $\mathbf{W}_{in}$ between all RCs.

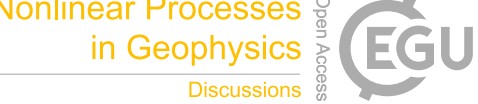

$n_{pack} * m_{pack}$ spatial points on the map. As there are $n * m$ total points on the map, the number of RCs needed will then be $P_f$ = $(n * m)/(n_{pack} * m_{pack})$. An illustration of the various packs on a sample map are provided in Fig. 1.

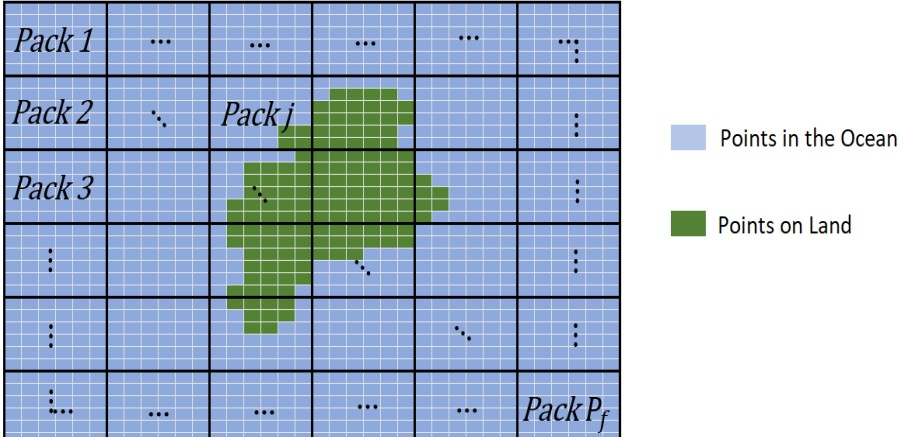

**Figure 1.** The various packs on a sample map. The points within a pack that are blue are the eligible points in the ocean that the pack's designated reservoir computer will attempt to forecast. Points in green represent land, which are not eligible points and whose indices are ignored in the formation of the pack.

No points containing land are assigned to a RC, therefore some packs will have more or less points than others, due to these points on land. It should also be noted that no point on the map can be in more than one pack, as packs do not cross. The sea surface temperatures of the points within the $j^{th}$ pack at the $i^{th}$ day are stacked in the vector $\boldsymbol{X}_i^{j_{pack}}$ which is of length $px$.

$$\boldsymbol{X}_i^{j_{pack}} = \begin{bmatrix} X_i^{j_{pack},1} \\ X_i^{j_{pack},2} \\ \vdots \\ X_i^{j_{pack},px} \end{bmatrix} \tag{4}$$

For the various reservoirs to interact with one another, coupling is introduced by finding the neighbors surrounding a pack. The neighbors of a pack are the non-land points that are either directly touching or on the corner of a point within the pack. As many points within a pack share similar neighbors, only unique neighbors are kept and their sea surface temperatures at the $i^{th}$ day are compiled into the vector $\boldsymbol{X}_i^{j_{neighbor}}$. The neighbors for a given pack on the sample map are illustrated below in Fig. 2.




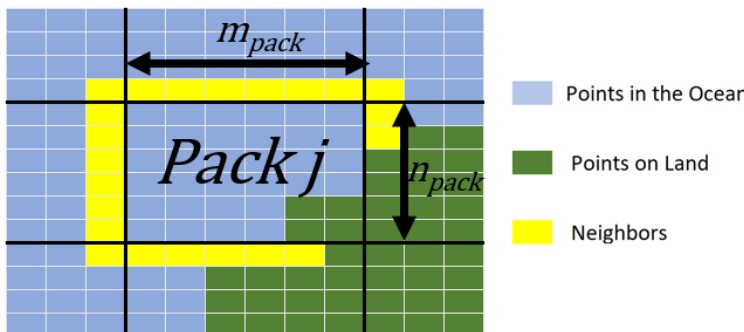

**Figure 2.** Illustration of the $j^{th}$ pack with it's valid neighbors denoted by a yellow highlight. In this example it is evident that $px$ is equal to 32 (number of pack points in the ocean) and $dx$ is equal to 54 ($px$ plus the number of neighbors). It is also evident that $n_{pack}$ and $m_{pack}$ are both equal to 6 in this case.

The vectors $\boldsymbol{X}_i^{j_{pack}}$ and $\boldsymbol{X}_i^{j_{neighbor}}$ are combined to form the vector $\boldsymbol{X}_i^{j}$ (of length $dx$), which contains all of the sea surface temperatures for the pack and it's neighbors at the $i^{th}$ day, and is ultimately the input for the $j^{th}$ reservoir computer.

$$
\boldsymbol{X}_i^{j} = \begin{bmatrix} \boldsymbol{X}_i^{j_{pack}} \\ \boldsymbol{X}_i^{j_{neighbors}} \end{bmatrix}
\tag{5}
$$

As we would like the model to predict sea surface temperatures within the pack at the next day $\boldsymbol{Y}_{i+1}^{j_{pack}}$, the reservoir computer for the $j^{th}$ pack can now be written as:

$\boldsymbol{r}_{i+1}^{j} = q(\mathbf{A}\boldsymbol{r}_i^{j} + \mathbf{W}_{in}^{dx}\boldsymbol{X}_i^{j} + \boldsymbol{b})$   (6)

$$
\boldsymbol{Y}_{i+1}^{j_{pack}} = \mathbf{W}_{out}^{j}\boldsymbol{r}_{i+1}^{j}
\tag{7}
$$

Each reservoir computer is trained over the entire training dataset from $i = 1 : t_{train}$ days. Each pack contains a distinct output matrix $\mathbf{W}_{out}^{j}$, such that the reservoir states are matched with the values of the SST within the pack at the next day. In order to save computer memory, it is advisable to create an array of input matrices with values of $dx$ from 1 to

$(n_{pack} + 2) * (m_{pack} + 2)$ and then assign these to RCs with a similar value of $dx$, denoted by $\mathbf{W}_{in}^{dx}$. One middle weight matrix $\mathbf{A}$ is shared between all the RCs, as the reservoir dimension $N$ is set to be fixed between all of the reservoirs. The architecture of a single reservoir computer is described in Fig. 3.




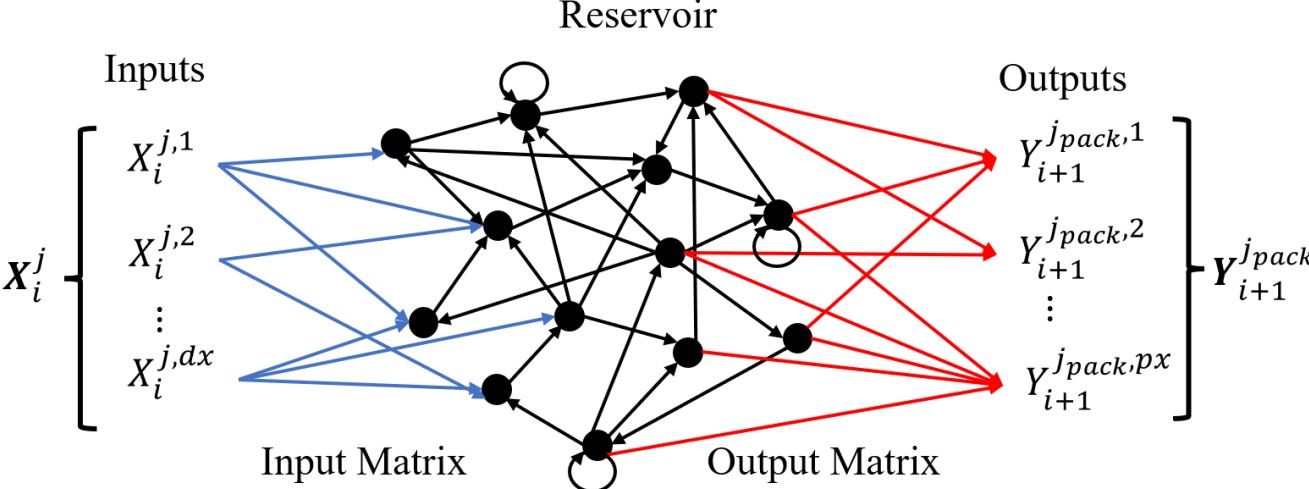

**Figure 3.** Illustration of the architecture of the $j^{th}$ reservoir computer.





## 4 Data

The dataset used to train and validate the model is titled "GHRSST Level 4 MUR 0.25deg Global Foundation Sea Surface
Temperature Analysis (v4.2)" which contains sea surface temperature data in degrees Kelvin on a global $0.25°$ grid from 2002
to 2021 in one day increments. This version is based on nighttime GHRSST L2P skin and sub skin SST observations from
several instruments, and is publicly available online via PODAAC (dat). The data was downloaded with the use of OPeNDAP
on 10/10/2021.

The years 2003 to 2020 of the data set were selected to form the training and validation data set. The data is given in an
equirectangular format, which is used throughout the modelling process for simplification purposes even though this conse-
quentially leads to a more refined mesh near the poles. The time series data was split into a training and validation data set,
consisting of 6,533 days and 42 days respectively.

In order to reduce the number of spatial points within the data set, the data was discretized such that the sea surface tem-
perature was now on a global $1.5°$ degree grid. This was performed by grouping original data points in a $6 \times 6$ matrix and
then taking the average over the group. If a grouping contained a point on land, this value was ignored in the computation of
the average of the group. Therefore, the data set for a given day went from a $720 \times 1440$ to a $120 \times 240$. Hence, $n = 120$ and
$m = 240$.

## 5 Forecasting

To subsequently forecast the global SST with the trained model, two different prediction types are performed. To begin testing
the short-term accuracy of the model, daily predictions are performed over the course of 6 weeks. The actual values of the SST
are fed as inputs into the reservoir during this time period and the predicted values for the next day are read out. Then, to test
the long-term accuracy of the model, the model is then allowed to run autonomously over the same 6 weeks. Now, the model
is still predicting SST each day but only has access to it's own previous prediction.

For both prediction types, the reservoir states are all cleared to zero prior to forecasting, and then ran over the 7 days prior
to the validation time frame, hence providing an initial condition for the model to begin from. The metaparameters that were
found to work well with the model are described in Table 1. The spectral radius $\rho$ of $\mathbf{A}$ was set to one by dividing $\mathbf{A}$ by it's
largest eigenvalue. One of the most important values in tuning the performance of the reservoir computer was found to be $\sigma$,
the value which determined the magnitude of the values in the input matrix $W_{in}$.

The model was implemented on the MATLAB R2019b platform, from a personal laptop, and the total training time was
slightly under 30 minutes. Even though the model predicts SST across the entire Earth, several time series at points chosen
arbitrarily are included in each section to compare the forecast to the actual SST over the validation time frame, the coordi-
nates of which can be found below in Table 2. It is likely, given the embarrassingly parallelizable nature of this task, that an
implementation that leverages GPU style computation could speed this stage up considerably.

To determine the quality of the forecast over the entire ocean, the mean absolute error (MAE), the root mean square error
(RMSE), and the maximum error in the forecast for a given day across the entire map are found. To find the MAE in the





**Table 1.** Metaparameters Used.

| Metaparameter | Value |
|---|---|
| $\sigma$ | $3e-4$ |
| $\rho$ | $1.0$ |
| $b$ | $0$ |
| $q(s)$ | $tanh(s)$ |
| $\lambda$ | $0.02$ |
| $d$ | $0.05$ |
| $N$ | $550$ |
| $n_{pack}$ | $4$ |
| $m_{pack}$ | $4$ |
| $t_{train}$ | $6,533$ days |
| $t_{validate}$ | $42$ days |

**Table 2.** Coordinates of Chosen Locations.

| Location | Latitude ($^\circ$ N) | Longitude ($^\circ$ E) |
|---|---|---|
| Cook Strait | $-41.25$ | $174.50$ |
| Gulf of Mexico near Key West, FL | $24.75$ | $-81.75$ |
| Coast of Gabon | $0.75$ | $8.25$ |
| Southern Ocean near Heard Island | $-55.00$ | $73.50$ |
| East Coast of Japan | $35.25$ | $141.75$ |
| Mozambique Channel | $-18.75$ | $41.25$ |
| Pacific Ocean near Tuvalu | $-8.25$ | $179.25$ |
| Coast of Ecuador | $-3.75$ | $-81.75$ |

forecast at a given day, a weighted average is performed on the error $e_i$ across the map, where $e_{p,i}$ denotes the absolute error at point $p$ on the map at the $i^{th}$ day. We perform this weighting due to the mesh being more refined near the poles compared to points near the equator, hence the area enclosed by each index $\Omega_p$ isn't constant. The actual area encompassed by a given point was found by simply using MATLAB's built in function areaquad(). The MAE in the forecast across the map at the $i^{th}$ day is then given by Equation 8, where $k$ is the number of points on the map that lie in the ocean ($k \approx 0.71 * n * m$).

$$MAE_i = \frac{1}{\sum_{p=1}^{k} \Omega_p} \sum_{p=1}^{k} e_{p,i}\Omega_p \qquad (8)$$

Similarly, the RMSE on the $i^{th}$ day is then given by Equation 9:

$$RMSE_i = \sqrt{\frac{1}{\sum_{p=1}^{k} \Omega_p} \sum_{p=1}^{k} e_{p,i}^2\Omega_p} \qquad (9)$$


Finally, the maximum error is simply the largest error in the forecast across all points on the $i^{th}$ day. To observe the effect of
the randomly selected input and middle weights on the performance of the RCs, the model was ran 15 times all with the same
metaparameters as described in Table 1, to collect data for the examination of the error.

### 5.1 Daily Forecasts

Daily forecasting operates by continually inserting the previous days actual SST $X_i$ into the reservoirs and then reading out
what the model predicts will be the SST at the next day $Y_{i+1}$, and then repeating this procedure over the course of the validation
time frame. The time series for the forecasted SST at the eight different points are provided below in Fig. 4 and is also matched
with the actual SST each day in Fig. 5.

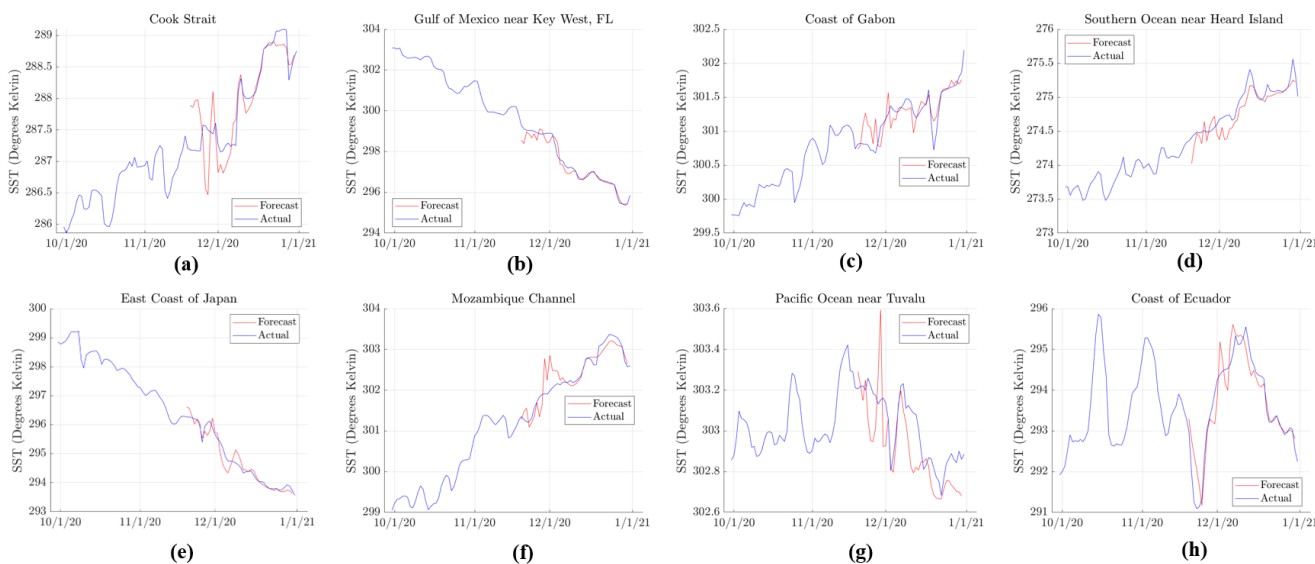

**Figure 4.** One day forecasts of the sea surface temperatures at various points in the ocean. The forecasted and the true SST is represented by
the red and the blue line respectively. The following locations are depicted in each figure: (a) Cook Strait, (b) the Gulf of Mexico near Key
West FL, (c) the Coast of Gabon, (d) the Southern Ocean near Heard Island, (e) the East Coast of Japan, (f) Mozambique Channel, (g) the
Pacific Ocean near Tuvalu, and (h) the Coast of Ecuador.



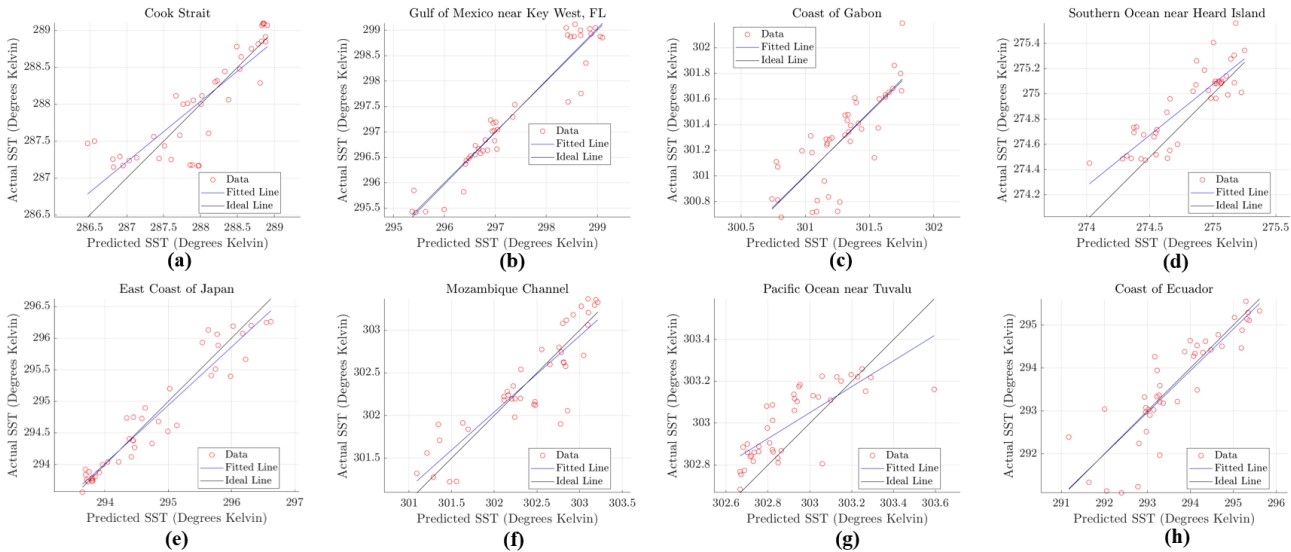

**Figure 5.** The true SST compared to the predicted SST for the model predicting one day at a time. The blue line indicates the resulting linear fit to the data, while the black line is the ideal place where the forecasted SSTs would match the true SST. The following locations are depicted in each figure: (a) Cook Strait, (b) the Gulf of Mexico near Key West FL, (c) the Coast of Gabon, (d) the Southern Ocean near Heard Island, (e) the East Coast of Japan, (f) Mozambique Channel, (g) the Pacific Ocean near Tuvalu, and (h) the Coast of Ecuador.

The corresponding regression statistics for the plots in Fig. 5 are described below in Table 3. Note that ideally, the slope $m$ would be equal to one, the intercept $b$ would be equal to zero, and the correlation coefficient $r$ would be one.

**Table 3.** Regression Statistics for the Daily Forecasts.

| Location | Slope $m$ | Intercept $b$ | Correlation Coefficient $r$ |
| --- | --- | --- | --- |
| Cook Strait | 0.82 | 51.76 | 0.85 |
| Gulf of Mexico near Key West, FL | 1.02 | 5.97 | 0.96 |
| Coast of Gabon | 0.98 | 6.99 | 0.79 |
| Southern Ocean near Heard Island | 0.81 | 53.17 | 0.86 |
| East Coast of Japan | 0.92 | 23.20 | 0.96 |
| Mozambique Channel | 0.89 | 32.73 | 0.89 |
| Pacific Ocean near Tuvalu | 0.62 | 115.17 | 0.78 |
| Coast of Ecuador | 0.97 | 8.45 | 0.87 |

For the daily forecasts, the correlation coefficients for the chosen sites are all 0.78 or greater, indicating a general relation
between the model's forecast and the true values at specific sites. The MAE, RMSE, and maximum error each daily forecast are described below in Fig. 6-8.


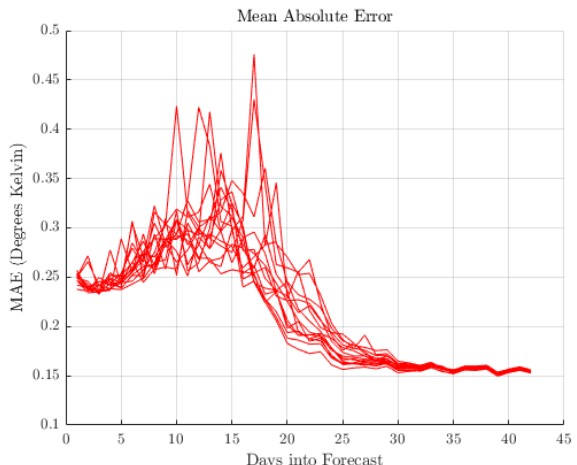

**Figure 6.** The mean absolute error for the one day forecast.

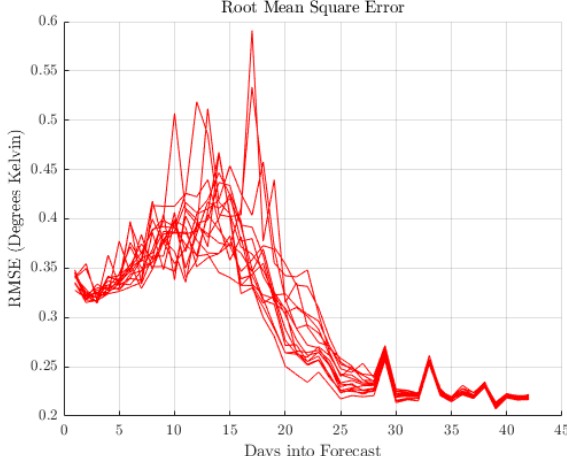

**Figure 7.** The root mean square error for the one day forecast.





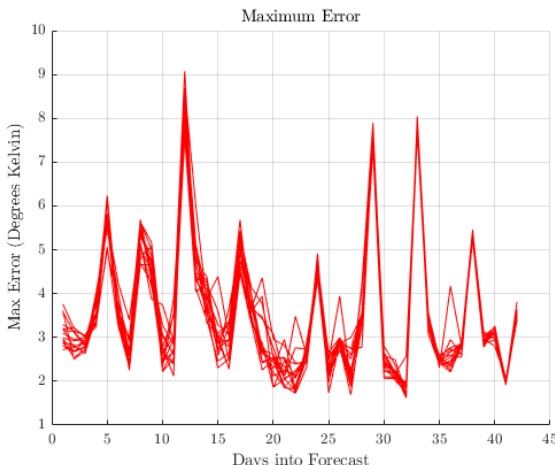

**Figure 8.** The maximum error for the one day forecast.

The MAE and the RMSE are initially 0.25 K and 0.35 K respectively, with all models experiencing a period of fluctuation in the error in their forecast between weeks 2 to 4, before settling down at the end of the forecast to a MAE and a RMSE of 0.15 K and 0.22 K respectively. It is an interesting feature that the error in the forecast decreases between the first and last day of forecasting, which is likely attributable to the reservoir state becoming enumerated with a greater number of previous SST values, hence gaining more knowledge about the previous states of the system.

## 5.2 6 Week Forecast

Meanwhile for the 6 week forecast, the forecasted sea surface temperatures $\boldsymbol{Y}_{i+1}$ are inputted back into the reservoir on the next day, therefore taking the place of the actual SST $\boldsymbol{X}_{i+1}$. This effectively allows the model to run autonomously over the validation time time frame for a total of 42 days. The eight time series for the forecasted SST are provided below in Fig. 9 and the data same data is also depicted on a day-wise basis in Fig. 10.



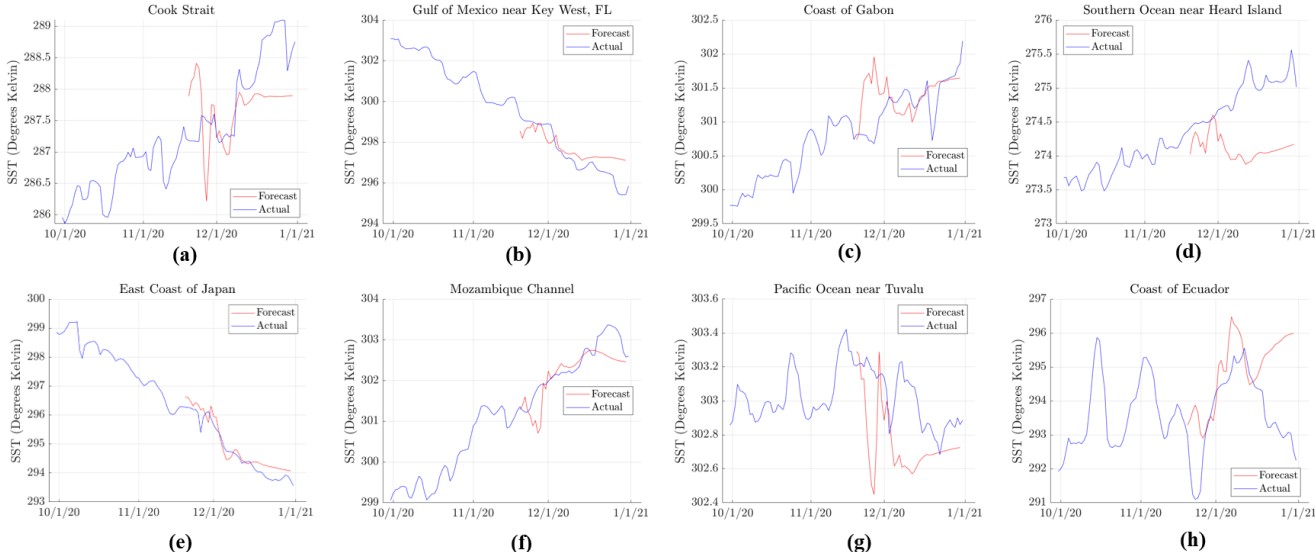

**Figure 9.** Forecasted sea surface temperatures at various points in the ocean with the model running autonomously. The forecasted and the actual SST is represented by the red and the blue line respectively. The following locations are depicted in each figure: (a) Cook Strait, (b) the Gulf of Mexico near Key West FL, (c) the Coast of Gabon, (d) the Southern Ocean near Heard Island, (e) the East Coast of Japan, (f) Mozambique Channel, (g) the Pacific Ocean near Tuvalu, and (h) the Coast of Ecuador.



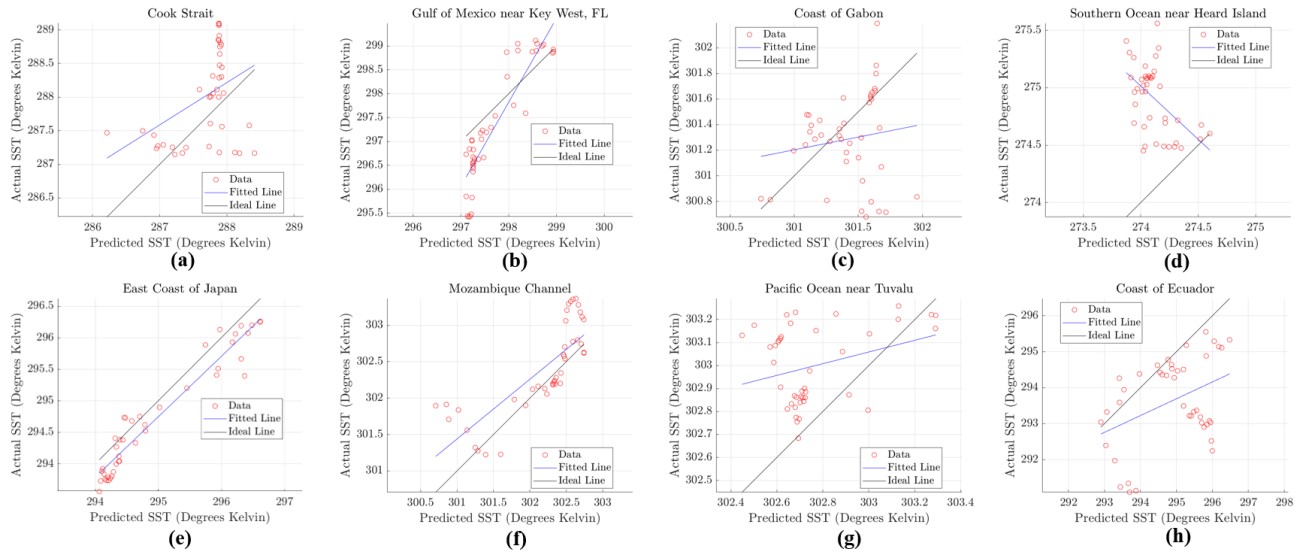

**Figure 10.** The actual SST compared to the predicted SST for the model running autonomously. The blue line indicates the resulting linear fit to the data, while the black line is the ideal place where the forecasted SSTs would match the actual SST. The following locations are depicted in each figure: (a) Cook Strait, (b) the Gulf of Mexico near Key West FL, (c) the Coast of Gabon, (d) the Southern Ocean near Heard Island, (e) the East Coast of Japan, (f) Mozambique Channel, (g) the Pacific Ocean near Tuvalu, and (h) the Coast of Ecuador.

The corresponding regression statistics for the plots in Fig. 10 are now described below in Table 4.

**Table 4.** Regression Statistics for the 6 Week Forecast.

| Location | Slope $m$ | Intercept $b$ | Correlation Coefficient $r$ |
|---|---|---|---|
| Cook Strait | 0.63 | 108.83 | 0.40 |
| Gulf of Mexico near Key West, FL | 1.76 | $-226.62$ | 0.91 |
| Coast of Gabon | 0.199 | 241.45 | 0.15 |
| Southern Ocean near Heard Island | $-0.93$ | 529.44 | $-0.52$ |
| East Coast of Japan | 0.96 | 12.22 | 0.96 |
| Mozambique Channel | 0.82 | 54.69 | 0.79 |
| Pacific Ocean near Tuvalu | 0.25 | 225.87 | 0.31 |
| Coast of Ecuador | 0.47 | 155.84 | 0.40 |

For the autonomous 6 week forecast, the correlation coefficients vary from -0.52 to 0.96, indicating that the model has an easier time predicting some locations than others. The MAE, RMSE, and maximum error in the forecast for each day the model
is forecasting are described below in Fig. 11-13.




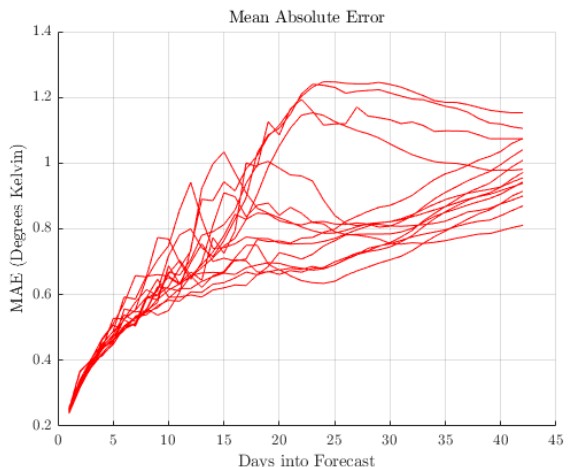

**Figure 11.** The mean absolute error for the 6 week forecast.

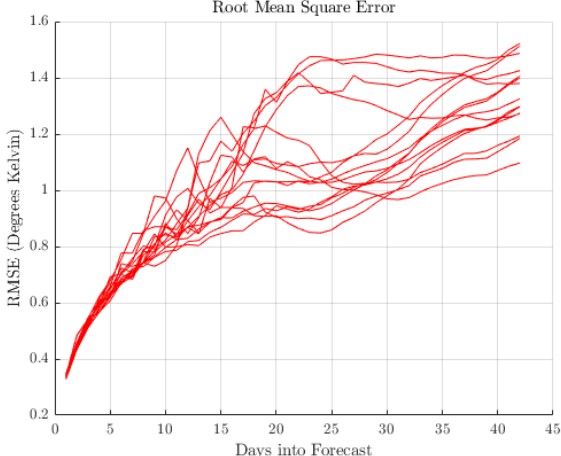

**Figure 12.** The root mean square error for the 6 week forecast.





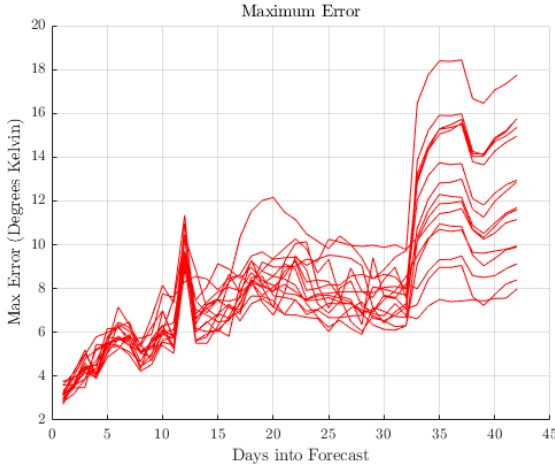

**Figure 13.** The maximum error for the 6 week forecast.

The MAE and the RMSE rises to 0.6 K and 0.8 K within the first week, which further increases as the model progresses. The maximum error in the forecast for all models is beneath 12 K for the first 4 weeks, and then the values begin to diverge depending on the model starting on the 32$^{nd}$ day. The general coherence in the relation between the error and the forecasting period is indicative that the randomness in the input and middle weights does slightly effect the model performance, but not drastically.

## 6 Conclusions

With the use of coupled reservoir computers, and specifically a collection of patches that represent local regions and designed to accommodate coastal-land interface variations, we were able to model for excellent forecasting the spatiotemporally complex dynamics of the global sea surface temperature over several weeks. The relative simplicity of the network architecture and the minimal training time is striking relative to other machine learning concepts. Even though our model is intended to describe the dynamics of the entire ocean, it is still able to predict SST at specific locations. In the future, it is of interest to explore the use of Next-Generation Reservoir Computers (NG-RC) in the task of predicting SST, as NG-RCs provide the added benefit of less metaparameters to tune compared to a traditional RC (Jaeger and Haas, 2004; Bollt, 2021b, a; Gauthier et al., 2021). It is also of interest to input other variables into the reservoir besides the SST, such as the surrounding air temperature (Jahanbakht et al., 2021) to observe if the results can be further improved.

*Code availability.* The code is available online at https://github.com/BenWalleshauser/Predicting-SST-w-.-Coupled-RCs.





*Author contributions.* The idea was originally conceived of by E.M.B. and B.W. was responsible for the creation of the model. B.W. and E.M.B. both contributed to the analyses and the creation of the paper.

*Competing interests.* The authors declare that they have no conflict of interest.

*Acknowledgements.* E.M.B. has received funding from the Army Research Office (ARO), the Defense Advanced Research Projects Agency (DARPA), the National Science Foundation (NSF) and National Institutes of Health (NIH) CRNS program, and the Office of Naval Research (ONR) during the period of this work.



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
