# Peer review of "Predicting Sea Surface Temperatures with Coupled Reservoir Computers"

_Nonlinear Processes in Geophysics, 2022_

## Author Comment (AC1)

**Predicting Sea Surface Temperatures with Coupled Reservoir Computers**

By: Ben Walleshauser and Erik Bollt

**Reviewer #1:**

The draft manuscript titled "Predicting Sea Surface Temperatures with Coupled Reservoir Computers" is an excellent effort in using coupled reservoir computers for predicting global sea surface temperatures. The manuscript may be accepted after minor revision after the following comments have been addressed:

Thank you for offering suggestions and pointing to areas of improvement for the manuscript. We believe we have adequately met these concerns and have revised the manuscript to better highlight the results and application of our model.

1.) Line 125 says that the actual values of SST were used for training. Can the authors comment on why normalization was not used as it has been shown to be necessary to train machine learning models?

Thank you very much for the comment. There are effectively two reasons we choose not to normalize the data. The first is that the reservoir state is scale-less, with scale being reintroduced by the application of the output matrix. The second is that our data source is univariate. We have included this point on lines 117-118 in the *Data* section, which is copied below for convenience.

Lines 117-118: We choose not to normalize the data, as the data is univariate and the reservoir state is effectively scale free, with scale being re-introduced with the trained output matrix.

2.) The authors train on a daily SST dataset, now oceans are known to operate on long temporal scales having a memory of at least a month. Can the authors comment on the utility of these forecasts? For example, Nino3, or Nino3.4 are considered based on monthly datasets precisely for the reason that the oceanic processes are slow.

This is a great point which we inadequately covered in the original manuscript. With regard to the daily forecasts, they can be used to fill in SST datasets when there is cloud coverage or data corruption, as one could initialize the model on the previous day and read out it's prediction for what would have been the SST on the missing day (Case, Jonathan L. et al., 2008). Then with regard to the extended 6 week forecast, this would be more applicable to near future weather prediction, as it would provide SST forecasts which could potentially be coupled with an atmospheric model. We have mentioned both of these applications in the *Forecasting* section of the revised manuscript, on lines 127-129 and 131-132 respectively, which are both copied below for convenience.

Lines 127-129: This type of forecast has a real world application in the form of filling in SST datasets when there is cloud cover or data corruption (Case, Jonathan L. et al., 2008), as the model can be used to estimate data for the missing days.

Lines 131-132: This form of the forecast would be more applicable to weather prediction, as it could be coupled with an atmospheric model to help predict near future weather patterns.

3.) Was any hyperparameter tuning performed?

Thank you for this concern, hyperparameter tuning was indeed performed via cross validation. It was found that the value of  $\sigma$  (the multiplier of the input matrix) produced the best results when it had an order of magnitude of  $10^{-4}$ . It was also observed that the results were enhanced with an increasing reservoir dimension N, though we decided to ultimately cap it at 1000 in the revised manuscript due to the associated increase in training time. We also explored the effect of the warmup time as suggested by RC2, and found that the model performed the best when ran for the 35 days prior to forecasting, with more days added not producing a noticeable increase in performance. The rest of the hyperparameters were not rigorously tuned, but rather chosen based off heuristics. We have included a very similar comment in lines 135-140 in the *Forecasting* section of the revised manuscript, which is also included below.

Lines 135-140: Via cross validation, several metaparameters ( $\sigma$ , N, and  $t_{warmup}$ ) were optimized and the values that were found to perform the best are described in Table 1. It was noticed that the results were not significantly increased for a value of  $t_{warmup}$  greater than 35 days. Results did consistently improve with an increasing reservoir dimension N, leading us to choose the value N = 1000. One of the more sensitive metaparameters was the value of  $\sigma$ , which was found to provide optimal results when having a magnitude of the order  $10^{-4}$ . In the spirit of simplicity, we choose not to rigorously optimize the remaining metaparameters, and rather choose them based off of heuristics.

4.) From figure 4, it can be seen that the model performs well where there is an established trend. For example, Fig 4a-f have a clear trend and the model is performing very well in all of them. There are some deviations in Fig 4g whereas Fig 4h is showing good performance. The intent of using such as coupled reservoir computer is to simulate the chaoticity of the system, whereas out of the 8 subplots in Figure 4, 6 have a clear linear trend where the model performs well, whereas in Fig 4g where there are some deviations, the model is not performing good relative to the previous subplots. Fig 4h is satisfactory. Can the authors maybe provide some more examples or describe these features from the results?

Thank you for this point, we have included two new locations (Bass Strait and the Laccadive Sea) in the revised manuscript which have nonlinear trends over the forecasting period. We have also included greater discussion of the time series trends at the 10 different locations in the revised manuscript for both lead times.

**Reviewer #2:**

A domain-decomposed set of coupled reservoirs that share the same set of hyperparameters are trained over multiple years of reanalysis daily sea surface temperature (SST) data and used to predict SST at lead times of between one day and six weeks.

The application of RC to the specific problem the authors consider, the analysis of the results and the write-up, all seem to be of a somewhat preliminary nature. As such it is not clear what a potential reader is expected to take away from this article. This issue needs to be addressed in a substantive fashion to be further considered for publication. A few other issues are noted below.

We immensely appreciate this comment, and we have taken several steps to make the paper more complete. This includes several comparisons to other models performing at similar lead times, a greater discussion of the effect of the model's intrinsic randomness on results, and a lengthened discussion following the results for both lead times. We have also included several more locations as recommended by RC1 and have redone part of the analysis for the 6 week forecast to better reflect the efficacy of the model over the time frame. Overall, we believe these revisions have made the results of the model much more communicable.

 The authors state: To observe the effect of the randomly selected input and middle weights on the performance of the RCs, the model was ran 15 times all with the same metaparameters as described in Table 1, to collect data for the examination of the error. Please state how the ensemble of predictions that use random variations of input and middle weights was analyzed. Meaning, what error is being shown is say figures 4-13

Thank you for this comment, as it pointed towards a deficiency in the original manuscript with respect to describing the global error statistics. In the original manuscript, each plot described how an individual model's error (evaluated over the entire globe) evolved over the forecasting period of 6 weeks. Therefore, on say forecasting day 7: the MAE, RMSE, and maximum error in an individual model's forecast are found by comparing it's forecast across the entire globe to the observed SST, and this process is subsequently repeated for the 15 other models (which have different input and middle weights) therefore leading to the 15 data points for day 7. In the revised manuscript, we have revised those figures to consist of an average and a standard deviation, and have added a new comment (lines 161-163) all in a manner which we believe makes the results more communicable to a potential reader.

Lines 161-163: These error values are found for each of the 15 models every day in the forecasting period, and then subsequently average values and standard deviations between models are found.

2.) In the context of Figs. 6 and 7, while the authors chose to spinup/warmup the reservoirs over a period of a week, the results suggest that a longer spinup of the reservoirs (of about 4 weeks) is called for. Please comment on the a priori choice of one week and the longer timescale that is required as indicated by the results. How does the longer timescale vary with changes in the hyperparameters?

We immensely appreciate this comment as it pointed out a substantial flaw in our original usage of the model. We have observed the effect of the warmup time on the model's results and have found that results are generally improved up until a warmup time of 35 days. With this new value, results are significantly improved for both 1 day and 6 week lead times. We have included a comment of this feature on line 136 and lines 180-182 in the *Forecasting* section of the revised manuscript, and have created a new hyperparameter titled  $t_{warmup}$ . Thank you again for this insight, as you have helped noticeably improve our results.

Line 136: It was noticed that the results were not significantly increased for a value of  $t_{warmup}$  greater than 35 days.

Lines 180-182: It should also be noted that our error values typically don't decrease over the forecasting period, indicating that the chosen warm-up time of 35 days is sufficient, as there would be a decline in the error over time if the reservoir was gradually benefiting from more provided information.

3.) What is the relevance of the leakage parameter in the context of a leaky reservoir update in this context? (and which the authors do not consider)

Thank you for your drawing attention to the fact that the RC could be further specialized to improve forecasting results. We believe that the new results presented in the revised manuscript are already very good, and elect not to optimize several metaparameters such as the leakage parameter, the spectral radius, and the level of sparsity in the spirit of simplicity. We do include a comment on the several metaparameters that we did optimize (the multiplier of the random input matrix, the reservoir dimension, and the warm up time) on lines 135-140 in the *Forecasting* section of the revised manuscript.

4.) Please comment on possible reasons for 4-5 day timescale seen in Figure 8, particularly since the data itself, e.g., as seen in fig. 4, seems to display variations on a broader range of timescales.

Thank you for this comment, as it pointed towards the fact that we needed to revise our axis labels in the time series figures to better reflect the results. In the revised manuscript, all of the figures depicting a time series (Fig. 4, 6, 7, and 9) are over the course of the forecasting period of 42 days (or greater to reflect the change in SST during the time leading up to the forecast), and we have included x-ticks in the same dd/mm/yyyy style for all figures rather than switching between that and numeric values.

5.) Given the results presented, it may seem somewhat of an over-statement when the authors state that "The results are demonstrated to replicate the actual dynamics of the system over a forecasting period of several weeks."

We greatly appreciate the honesty of this comment and have revised the statement in the revised manuscript where the word "replicate" is replaced with "generally follow."

References:

Case, Jonathan L., Santos, Pablo, Lazarus, Steven M., Splitt, Michael E., Haines, Stephanie L., Dembek, Scott R., and Lapenta, William M.: A Multi-Season Study of the Effects of MODIS Sea-Surface Temperatures on Operational WRF Forecasts at NWS Miami, FL, New Orleans, LA, 2008.